# Genomics of an endemic cystic fibrosis *Burkholderia multivorans* strain reveals low within-patient evolution but high between-patient diversity

Cédric Lood[1,2‡], Charlotte Peeters[3‡], Quentin Lamy-Besnier[1], Jeroen Wagemans[1], Daniel De Vos[4], Marijke Proesmans[5], Jean-Paul Pirnay[4], Fedoua Echahidi[6], Denis Piérard[6], Matthieu Thimmesch[7], Anca Boeras[8], Katrien Lagrou[9,10], Evelien De Canck[3], Elke De Wachter[11], Vera van Noort[2,12], Rob Lavigne[1]*, Peter Vandamme[3]*

1 Department of Biosystems, Laboratory of Gene Technology, KU Leuven, Leuven, Belgium, 2 Department of Microbial and Molecular Systems, Centre of Microbial and Plant Genetics, Laboratory of Computational Systems Biology, KU Leuven, Leuven, Belgium, 3 Belgian National Reference Centre for *Burkholderia*, Laboratory of Microbiology, Department of Biochemistry and Microbiology, Faculty of Sciences, Ghent University, Ghent, Belgium, 4 Laboratory for Molecular and Cellular Technology (LabMCT), Queen Astrid Military Hospital, Brussels, Belgium, 5 Department of Pediatrics, University Hospital Leuven, University of Leuven, Leuven, Belgium, 6 Belgian National Reference Centre for *Burkholderia*, Department of Microbiology, Universitair Ziekenhuis Brussel, Vrije Universiteit Brussel (VUB), Brussels, Belgium, 7 Department of Pediatric Pneumology, CHC MontLégia, Liège, Belgium, 8 Department of Microbiology, CHC MontLégia, Liège, Belgique, 9 Department of Microbiology, Immunology and Transplantation, KU Leuven, Leuven, Belgium, 10 Clinical department of Laboratory Medicine, University Hospital Leuven, Leuven, Belgium, 11 Department of Pediatric Pulmonology, Universitair Ziekenhuis Brussel, Vrije Universiteit Brussel (VUB), Brussels, Belgium, 12 Institute of Biology, Leiden University, Leiden, The Netherlands

‡ These authors are joint first authors on this work.
* rob.lavigne@kuleuven.be (RL); peter.vandamme@ugent.be (PV)

**Data Availability Statement:** All genomics data was deposited in the NCBI GenBank database. The initial Illumina and Nanopore sets of reads are

## Abstract

*Burkholderia multivorans* is a member of the *Burkholderia cepacia* complex (Bcc), notorious for its pathogenicity in persons with cystic fibrosis. Epidemiological surveillance suggests that patients predominantly acquire *B. multivorans* from environmental sources, with rare cases of patient-to-patient transmission. Here we report on the genomic analysis of thirteen isolates from an endemic *B. multivorans* strain infecting four cystic fibrosis patients treated in different pediatric cystic fibrosis centers in Belgium, with no evidence of cross-infection. All isolates share an identical sequence type (ST-742) but whole genome analysis shows that they exhibit peculiar patterns of genomic diversity between patients. By combining short and long reads sequencing technologies, we highlight key differences in terms of small nucleotide polymorphisms indicative of low rates of adaptive evolution within patient, and well-defined, hundred kbps-long segments of high enrichment in mutations between patients. In addition, we observed large structural genomic variations amongst the isolates which revealed different plasmid contents, active roles for transposase IS*3* and IS*5* in the deactivation of genes, and mobile prophage elements. Our study shows limited within-patient *B. multivorans* evolution and high between-patient strain diversity, indicating that an environmental microdiverse reservoir must be present for this endemic strain, in which

available in the SRA database via the accession numbers of Table A in S1 Tables. The genomes and their annotations can be found via the BioProject accession number PRJNA601875.

**Funding:** CL is supported by a PhD fellowship from FWO Vlaanderen (1S64720N). The labs of VvN and RL are supported by the ID-N grant PHAGEFORCE from the KU Leuven (IDN/20/024). Part of this work was performed in the framework of the Belgian National Reference Centre for Burkholderia (CP, PV, DP), supported by the Ministry of Social Affairs through a fund within the National Health Insurance System. The Oxford Genomics Centre at the Wellcome Centre for Human Genetics is funded by Wellcome Trust grant reference 203141/Z/16/Z. The funders had no role in study design, data collection and analysis, decision to publish, or preparation of the manuscript.

**Competing interests:** The authors have declared that no competing interests exist.

active diversification is taking place. Furthermore, our analysis also reveals a set of 30 parallel adaptations across multiple patients, indicating that the specific genomic background of a given strain may dictate the route of adaptation within the cystic fibrosis lung.

## Author summary

In many countries, *Burkholderia multivorans* is the most prevalent species within the *Burkholderia cepacia* complex (Bcc) found infecting the lungs of patients with cystic fibrosis (CF). Its positive identification is of immediate concern to the health of the patient as it is notoriously hard to eradicate using antibiotics and can cause necrosis of the lung tissues (cepacia syndrome). Infection control measures reduced the prevalence of *B. cenocepacia* in CF wards, but patients continue to acquire infections by *B. multivorans* from environmental sources. In most reported cases, the infecting strains are unique except in rare cases in which cross-infection is observed between patients. We report here an endemic strain of *B. multivorans* with sequence type ST-742 that has been infecting multiple patients, without evidence for cross-infection. We investigated the epidemiology and genomics of this ST-742 strain and show that it is microdiverse, as isolates between-patients exhibit numerous genomic differences, at scales that have not been observed previously when looking at evolutionary trajectories within-patients. Additionally, we found that the specific genomic background of a given strain may dictate the strategy of adaptation within the CF lung.

## Introduction

*Burkholderia cepacia* complex (Bcc) bacteria are relatively rare but notorious opportunistic pathogens in cystic fibrosis (CF) patients. They are associated with higher rates of morbidity and mortality, as well as lower post-lung transplant survival [1–4]. Bcc infections in CF are characterized by highly variable clinical outcomes, but commonly result in a progressive decline of lung function. In extreme cases, Bcc infection can result in "cepacia syndrome", a necrotizing pneumonia and septicemia that engages a lethal prognosis for the patient [5]. Bcc infections are difficult to eradicate because the infecting strains have an innate resistance to multiple antibiotics [3,4,6].

Infection control measures and patient segregation were globally implemented in the 1990s to reduce patient-to-patient transmission and thereby the prevalence of *Burkholderia cenocepacia* [7–10]. Subsequently, *Burkholderia multivorans* emerged as the most prevalent Bcc representative in many countries [1,2,4,11–13]. *B. multivorans* is considered less virulent than *B. cenocepacia* [6,8] and infections are often characterized as chronic with episodes of exacerbations [3,4]. Nevertheless, several cases of "cepacia syndrome" and *B. multivorans* epidemic outbreaks have been reported [14–17].

The ability to differentiate Bcc strains has been key in understanding their epidemiology and improving infection control guidelines for the CF community. Multilocus sequence typing (MLST) is a well-established molecular technique to study the epidemiology and population structure of Bcc organisms [18–20]. The Bcc MLST scheme is based on the allelic variations of seven housekeeping genes (*atpD*, *gltB*, *gyrB*, *recA*, *lepA*, *phaC* and *trpB*) and each strain is defined by its unique allelic profile and sequence type (ST) [18,21]. While several *B. multivorans* STs were shown to be globally distributed [22,23], only a limited number of genetically distinct *B. multivorans* outbreak strains have been described [22]. The small number of *B.*

*multivorans* outbreaks and the fact that *B. multivorans* isolates from CF patients mostly represent unique strains strongly suggest that person-to-person transmissions are limited and that *B. multivorans* strains are usually acquired from environmental sources [10,12,22,24,25].

*B. multivorans* has been the most prevalent Bcc species in Belgian CF patients for more than three decades [1,26,27]. *B. multivorans* isolates from Belgian CF patients typically represent unique strains, but rather unexpectedly, several *B. multivorans* isolates with the same sequence type (ST-742) were found in multiple Belgian CF patients treated in different CF centers [26,28]. A study was performed to investigate the epidemiology and risk factors that play a role in the spread of these clones, but no evidence to support cross-infection was detected [26], suggesting these patients acquired this *B. multivorans* strain from the natural environment.

The aim of the current study was to investigate the genomic epidemiology and evolution of *B. multivorans* ST-742 that were isolated from respiratory samples of four Belgian CF patients over a period of at least ten years. We combined sequencing technologies to enable an optimal resolution in the *de novo* reconstruction of the genomes and elucidation of structural variations [29,30], with clinical and antibiotic resistance data to discriminate between ST-742 isolates. We show that ST-742 isolates of different patients have distinct genomic profiles and propose that each patient has been colonized independently from a natural reservoir of ST-742 where recombination events diversify the population of this microdiverse endemic *B. multivorans* strain.

## Results

### Isolates from four different patients share the same sequence type

Between 2009 and 2019, thirteen *B. multivorans* isolates with ST-742 were obtained from respiratory samples of four unrelated patients attending three different pediatric CF centers in Belgium (Fig 1, Table 1, Table A in S1 Tables) [26].

### Variability of antibiotic resistance profiles in the ST-742 isolates

We tested the susceptibility of our isolates to 20 different antibiotics and interpreted as resistant, low level of resistance, or susceptible based on EUCAST interpretation of the Minimum Inhibitory Concentrations (MIC) value when available, or the species independent Pharmacokinetics/Pharmacodynamics (PK/PD) interpretation otherwise (Fig 2). The complete table with the MIC values is given in Table B in S1 Tables.

Isolates of the first positive cultures (i.e., P1Bm2009 and P3Bm2015) were susceptible to multiple antibiotics (Fig 2). Interestingly, P2Bm2011b, and P4Bm2019 also display similar profiles despite not being the first *B. multivorans* isolate for the respective patients. For patient 1, pairs of isolates from the same sample with distinct colony morphologies (P1Bm2011a-b, P1Bm2012a-b) showed highly similar or identical profiles; the last isolate (P1Bm2015) was resistant to all antibiotics tested, except trimethoprim/sulfamethoxazole. For patient 2, the two isolates from the same sample with distinct colony morphologies (P2Bm2011a-b) showed different profiles: one isolate (P2Bm2011b) had a resistance profile similar to that of other initial isolates (i.e. P1Bm2009 and P3Bm2015), while the other isolate (P2Bm2011a) had a resistance profile that resembled that of later isolates such as P1Bm2012b. The resistance profile of the isolate P2Bm2018 was most similar to that of P2Bm2011a and P1Bm2012b (Fig 2).

### The replicons are collinear and broadly share nucleotide sequence identity across isolates

The hybrid sequencing data, combining Nanopore and Illumina sequencing, enabled highly contiguous *de novo* reconstructions of the genomes for all thirteen isolates. The genomes

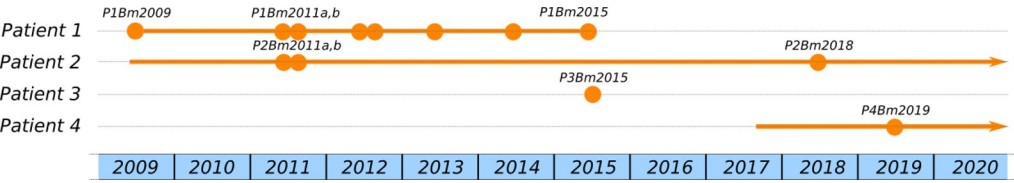

**Fig 1. Timeline of *B. multivorans* ST-742 isolations from sputum samples in Patients 1 to 4.** Each dot represents an isolate (exact isolation dates in Table 1). The 2011 and 2012 isolates of patient 1 and 2012 isolates of patient 2 each originated from the same sputum sample and were studied in parallel as they displayed distinct colony morphologies. The line connecting the isolates indicates the infection timeline, with the infections of patients 2 and 4 still currently ongoing.

ranged between 6.5 and 6.7 Mb in size with a G+C content of approximately 67%. The functional annotation revealed 5,837 to 6,045 coding DNA sequences (CDS) per genome (Table 1). The annotation of antibiotic resistance genes revealed a single class A beta-lactamase encoding gene that was present in all isolates and was a homolog to *penA*, previously identified in *B. cepacia* ATCC 25416 (NG_048030.1). Each of the replicons contained multiple genomic islands that included prophage elements and secondary metabolites (Table C in S1 Tables). No clustered regularly interspaced short palindromic repeats (CRISPRs) were found.

The assembly graphs (S1 Fig) confirmed the typical highly conserved genome structure of *B. multivorans* with three large replicons [6,23] (from here on referred to as C1, C2 and C3). The genomic sequences of each of the three chromosomes across all isolates were collinear (S2 Fig) and broadly shared nucleotide sequence identity (>99.8%).

## The *B. multivorans* population structure reveals a unique, clonal ST-742 cluster

There are no other reports of ST-742 isolates in the Bcc PubMLST database [31]. Furthermore, MLST analysis of all publicly available *B. multivorans* genomes (Table D in S1 Tables) did not

**Table 1. List of isolates included in our study.** Isolates marked with an asterisk were obtained from the same sputum sample but displayed distinct colony morphologies.

|  | Culture date | Size (bp) | CDS | NCBI Accession |
|---|---|---|---|---|
| **Patient 1 (˚1999)** |  |  |  |  |
| P1Bm2009 | 29/01/2009 | 6,569,085 | 5,917 | CP048460-63 |
| P1Bm2011a | 31/03/2011* | 6,498,339 | 5,830 | CP048557-59 |
| P1Bm2011b | 31/03/2011* | 6,531,926 | 5,829 | CP048454-56 |
| P1Bm2012a | 16/02/2012* | 6,545,859 | 5,828 | JAAEEF000000000 |
| P1Bm2012b | 16/02/2012* | 6,495,911 | 5,828 | CP048451-53 |
| P1Bm2013 | 25/07/2013 | 6,496,014 | 5,830 | JAAEEE000000000 |
| P1Bm2014 | 21/01/2014 | 6,497,010 | 5,831 | CP048448-50 |
| P1Bm2015 | 12/06/2015 | 6,499,180 | 5,833 | CP048445-47 |
| **Patient 2 (˚2004)** |  |  |  |  |
| P2Bm2011a | 11/07/2011* | 6,705,051 | 5,982 | CP048441-44 |
| P2Bm2011b | 11/07/2011* | 6,707,647 | 5,983 | JAAEED000000000 |
| P2Bm2018 | 20/08/2018 | 6,661,721 | 5,988 | JACKVR000000000 |
| **Patient 3 (˚2006)** |  |  |  |  |
| P3Bm2015 | 09/11/2015 | 6,725,592 | 6,045 | JAAEEC000000000 |
| **Patient 4 (˚1992)** |  |  |  |  |
| P4Bm2019 | 23/12/2019 | 6,538,969 | 5,799 | JACKVS000000000 |

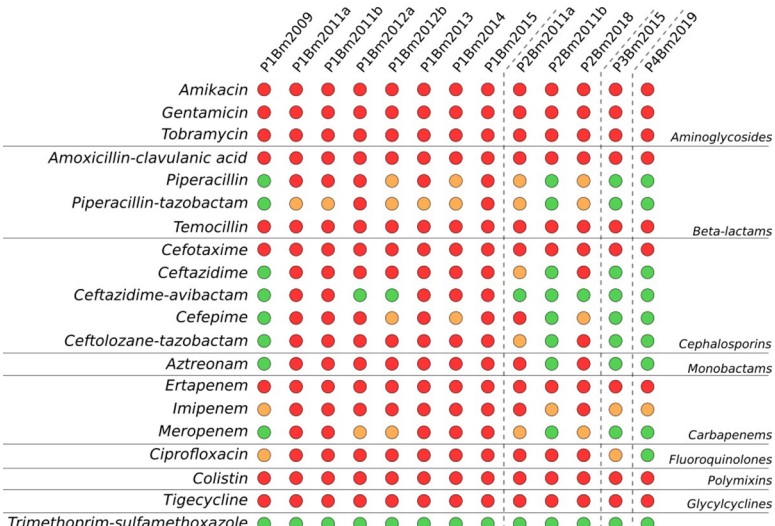

**Fig 2. Antibiotic resistance profiles of the thirteen isolates.** MIC values (Table B in S1 Tables) were interpreted using the EUCAST or PK/PD when available, and the results are summarized visually: Red = resistant, orange = low level of resistance, and green = sensitive.

reveal other ST-742 isolates either. To examine the position of ST-742 isolates in the general population structure of *B. multivorans* a core-genome based phylogenomic analysis was performed of the genomes of our thirteen ST-742 isolates together with 97 publicly available *B. multivorans* genomes (Table D in S1 Tables). The core genome consisted of 995 genes that were used to build a multiple sequence alignment and construct a core-genome phylogeny

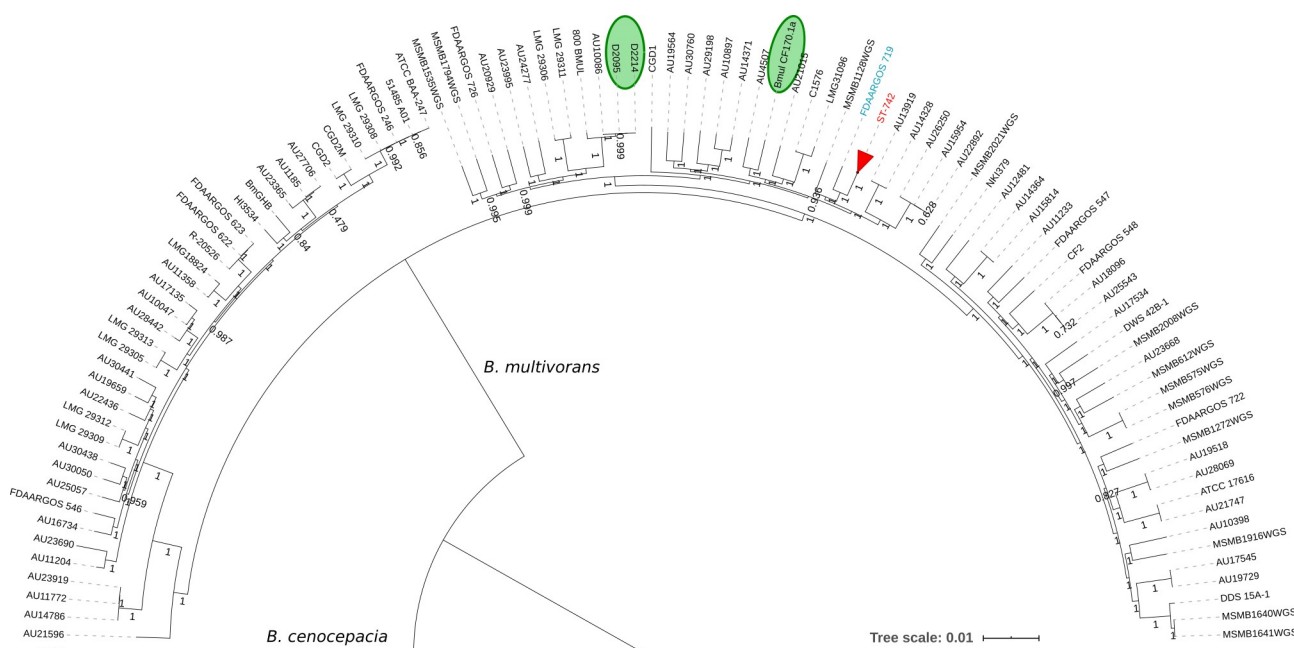

**Fig 3. Population structure of *Burkholderia multivorans*.** We established the population structure via the isolation of core genes across all publicly available *B. multivorans* genomes in order to position our cluster of isolates in the population. We collapsed the subtree (highlighted in red) consisting of our thirteen ST-742 isolates as they showed to be clonal within the population. *B. cenocepacia* J2315 was used as outgroup. The strains highlighted in green are from the study by Silva et al. (2016) and Caballero et al. (2018), D2095 with D2214 and Bmul_CF170.1a respectively.

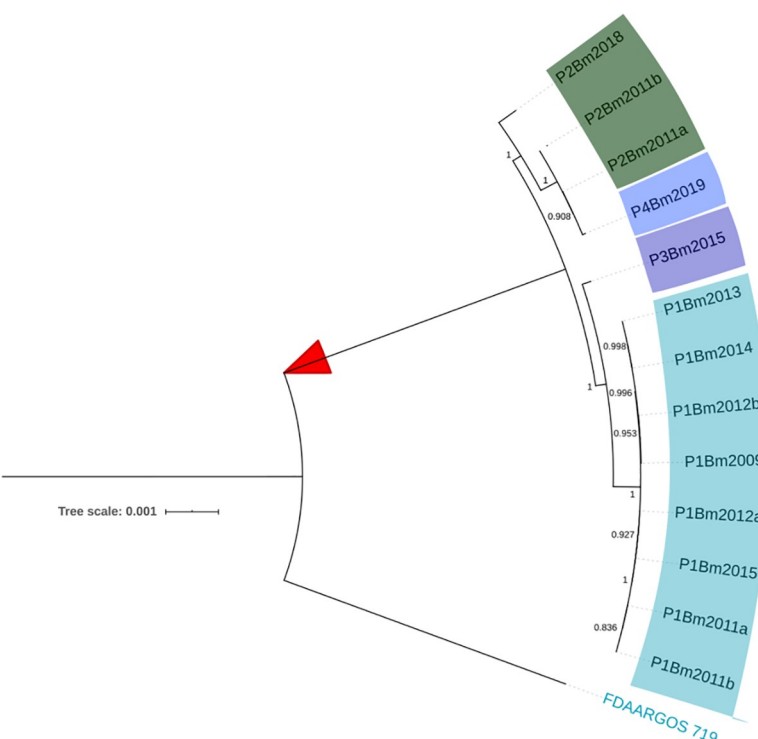

**Fig 4. Core-SNP phylogeny of the ST-742 isolates.** We used the information contained in the 61,845 core SNPs found between our cluster of ST-742 isolates and the strain FDAARGOS_719 to further zoom in on the evolutionary relations between the ST-742 isolates.

(Fig 3). This analysis demonstrated that ST-742 isolates represent a unique line of descent within *B. multivorans* with FDAARGOS_719, a clinical isolate from the US [32], as nearest neighbor.

To further investigate the evolutionary relationships within the ST-742 cluster, a SNP analysis was performed to detect small insertions or deletions (indels) and single nucleotide polymorphisms (SNPs) in the ST-742 genomes and the FDAARGOS_719 genome. The resulting core SNP alignment was used for a second, more detailed phylogenetic analysis (Fig 4). While all patient 1 isolates clustered closely together, this was not the case for the isolates from patient 2. Indeed, the core SNP analysis revealed that the two initial patient 2 isolates (P2Bm2011a-b) were more similar to each other and to the single patient 4 isolate (P4Bm2019), than to the later isolate of patient 2 (P2Bm2018) (Fig 4).

## Distinct plasmid content and large structural variations between patients

Plasmids could be detected in at least one isolate of each of the four patients but appears to be lost during chronic infection (patient 1 and 2). Pairwise comparisons of these four plasmid sequences using BLASTn revealed that the plasmid of patient 1 isolate P1Bm2009 had only 4% coverage with high identity (97%) to the group of plasmids found in patient 2, 3, and 4 (P2Bm2011a-b, P3Bm2015, and P4Bm2019). When compared against the NCBI nr/nt database [33], we found a best BLASTn hit (83% identity over 20% sequence coverage) to the pINT23 plasmid of *Burkholderia pseudomultivorans* SUB-INT23-BP2 (CP013379.1). On the other hand, the group of plasmids from the first isolates of patients 2 to 4 shared a pairwise sequence identity and coverage of over 99% and had a best database hit (99% identity and 66%

**Table 2. Large structural variations in the isolates of Patient 2–4.** By setting the isolate P1Bm2009 as a reference, we mapped the long Nanopore reads of the isolates from patients 2 to 4 to identify large structural variations that escape SNPs analysis. We omitted from this list indels found in intergenic regions (11) and hypothetical proteins (8). The full list of structural variations can be found in Tables G and H in S1 Tables.

| Strains | Chr | Position | Function |
|---|---|---|---|
| P2Bm2011b | 1 | 17536 | Insertion of IS3 into type IV secretion system tip protein VgrG |
| P2Bm2011b, P2Bm2018 | 1 | 59006 | Insertion of IS5 into intergenic promoter of type II secretion system secretin GspD |
| P2Bm2018 | 1 | 482440 | Deletion in long-chain fatty acid—CoA ligase |
| P4Bm2019 | 1 | 762540 | Deletion EamA family transporter, hypothetical prot, RES family NAD+ phosphorylase |
| P4Bm2019 | 1 | 825939 | Deletion glutamine—fructose-6-phosphate transaminase (isomerizing) |
| P2Bm2011b, P2Bm2018 | 1 | 860357 | Large deletion including regulator proteins |
| P2Bm2011b, P2Bm2018 | 1 | 1084705 | Deletion IS3 family transposase |
| P2Bm2011b | 1 | 1126373 | Insertion of IS3 into GlxA family transcriptional regulator |
| P2Bm2011b, P2Bm2018, P4Bm2019 | 1 | 1250973 | Deletion IS256 family transposase |
| P2Bm2011b | 1 | 1383320 | Insertion of IS5 into ornithine acetyltransferase |
| P2Bm2011a, P2Bm2018 | 1 | 2112493 | Prophage deletion |
| P2Bm2011a, P2Bm2018 | 1 | 2882469 | Insertion of IS5 into PTS sugar transporter/HPr kinase/phosphorylase |
| P2Bm2011a,b, P2Bm2018 | 2 | 199631 | Deletion in polyketide cyclase |
| P2Bm2011b, P2Bm2018 | 2 | 234973 | Insertion of IS5 into peroxidase-related enzyme |
| P2Bm2011b | 2 | 297796 | Insertion of IS5 into transposase |
| all | 2 | 386097 | Deletion IS3 family transposase |
| P2Bm2011b | 2 | 851128 | Insertion of IS5 into type IV secretion system baseplate subunit TssF |
| P2Bm2011b | 2 | 865794 | Insertion of IS3 into DUF3304 domain-containing protein |
| P2Bm2018 | 2 | 906456 | Insertion of IS5 into DEAD/DEAH box helicase |
| P2Bm2011b | 2 | 921221 | Insertion of IS5 into multidrug transporter subunit MdtN |
| P2Bm2011b | 2 | 1052623 | Insertion of IS5 into twin-arginine translocation pathway signal |
| P2Bm2011b, P2Bm2018 | 2 | 1770117 | Insertion of IS5 into Tat pathway signal protein |
| P2Bm2011b | 2 | 1892870 | Prophage deletion |
| P2Bm2011b | 2 | 2294168 | Insertion of IS5 into aldo/keto reductase |
| P2Bm2011a,b, P2Bm2018 | 3 | 128274 | Prophage deletion |
| P2Bm2011a,b | 3 | 599926 | Insertion of IS5 into PLP-dependent aminotransferase family protein |

sequence coverage) to the pTGL1 plasmid of *Burkholderia multivorans* ATCC 17616 (AP009388.1). The functional content of the plasmids found in P1Bm2009 and P2Bm2011a (taken as representative of the group of plasmids found in P2Bm2011a-b, P3Bm2015, and P4Bm2019) is described in Tables E and F of S1 Tables. The plasmids differ extensively in size (72 kbp for P1Bm2009 and 162 kbp for P2Bm2011a) and in functional content. Indeed, over 24% of the large plasmid genes (P2Bm2011a) are related to DNA maintenance and repair compared to 9% in the small plasmid (P1Bm2009) (Table F of S1 Tables). Both plasmids contain proteins linked to Type IV secretion systems, with the smaller plasmid (P1Bm2009) specifically carrying a VirB4 component. Importantly, we did not identify any antibiotics resistance associated genes on those plasmids.

The long-read sequencing data were further used to survey larger structural variations between isolates (Table 2, Tables G and H of S1 Tables). This analysis revealed an active role for transposons from families IS3 and IS5 in the deactivation of genes (Table 2), particularly in the isolates from patient 2. Additionally, three complete prophages were present in all isolates except those of patient 2, in which they appear to be actively mobile. Indeed, the prophage located on C3 was absent from all isolates of patient 2, but the prophage on C2 was absent in P2Bm2011b, while the prophage on C1 was absent in P2Bm2011a and P2Bm2018.

## ST-742 isolates have regions with unusually high SNP densities in C1 and C2

To assess in which regions of the chromosomes SNPs occurred, all identical SNPs found on the different genomes of isolates from patient 1 were collapsed into a single collection of SNPs (85 in total). The same analysis was performed on the isolates P2bm2011a and P2Bm2011b from patient 2 (4,694 in total). However, the later isolate of patient 2 (P2Bm2018) was kept separated as it displayed a large number of unique variants (1,990 SNPs when compared to P2Bm2011a-b, S3 Fig).

Figs 5, 6 and 7 visually summarize the SNP densities for each chromosome and sets of isolates. The full list of SNPs is provided in Table I in S1 Tables. S3 Fig shows the number of shared SNPs among the isolates.

In general, SNPs were randomly distributed across the chromosomes, with 53, 20 and 12 SNPs on C1, C2 and C3 of isolates from patient 1, respectively. However, an enrichment of SNPs was detected in three regions for the isolates from the other patients: a first centered around position 900 kbp for C1 (Fig 5), and a second and third in C2 around 170 kbp and 310 kbp, respectively (Fig 6). No such pattern was observed in the smaller C3 (Fig 7). On C1, this region of P2Bm2011a-b, P2Bm2018, P3Bm2015 and P4Bm2019 all ended at the exact same position (991,509) but started at different positions (702,401–794,204) (Table J in S1 Tables). On C2, the enrichment of SNPs was observed in all isolates except in isolate P3Bm2015 (Fig 6). These regions all start at the same position (150,841 and 318,125) but ended at different positions (193,566–204,452 and 331,399–331,929). The analysis of the functional content found in these regions did not reveal specific enrichments for certain COG categories when compared to the whole genomic background of the strain (Table M in S1 Tables).

## Genome evolution of patient 1 isolates over the course of the infection and parallel evolution in different patients

To reveal within-patient genomic variations at the level of SNPs and larger structural variations, the genomes of all sequential isolates of patient 1 were compared using isolate P1Bm2009 as a reference. When collapsing all SNPs as described above, a total of 85 unique SNPs were detected on the three chromosomes. Many of these mutations were fixed over time once they appeared (Fig 8, Table I in S1 Tables). Some genes had mutations at multiple independent positions (Fig 8, Table 3). Both pairs of isolates P1Bm2011a-b and P1Bm2012a-b, which originated from the same sputum samples but displayed a different colony morphology, also showed a distinct SNP profile. This suggests an evolution of different population lineages within this patient. These 85 SNPs occurred over six years, yielding an average mutation rate of 15 SNPs/year (S4 Fig).

Additionally, we searched for evidence of parallel evolution of loci across multiple patients. In total, 30 CDS (and ten non-coding regions) were detected with mutations in at least three patients, and only two loci (129 and 6144) had mutations in all patients (Table I in S1 Tables and Table 4).

Larger structural variations that were not detected using the SNP analysis included the loss of a plasmid that was only detected in the first isolate P1Bm2009 (see above), as well as several deletions and insertions, some of which could again be linked to transposase activity (Table 5, Table H in S1 Tables).

## Discussion

*B. multivorans* has been the most prevalent Bcc species in Belgian CF patients for more than three decades [1,26,27] and patients generally carry distinct sequence types, suggesting the

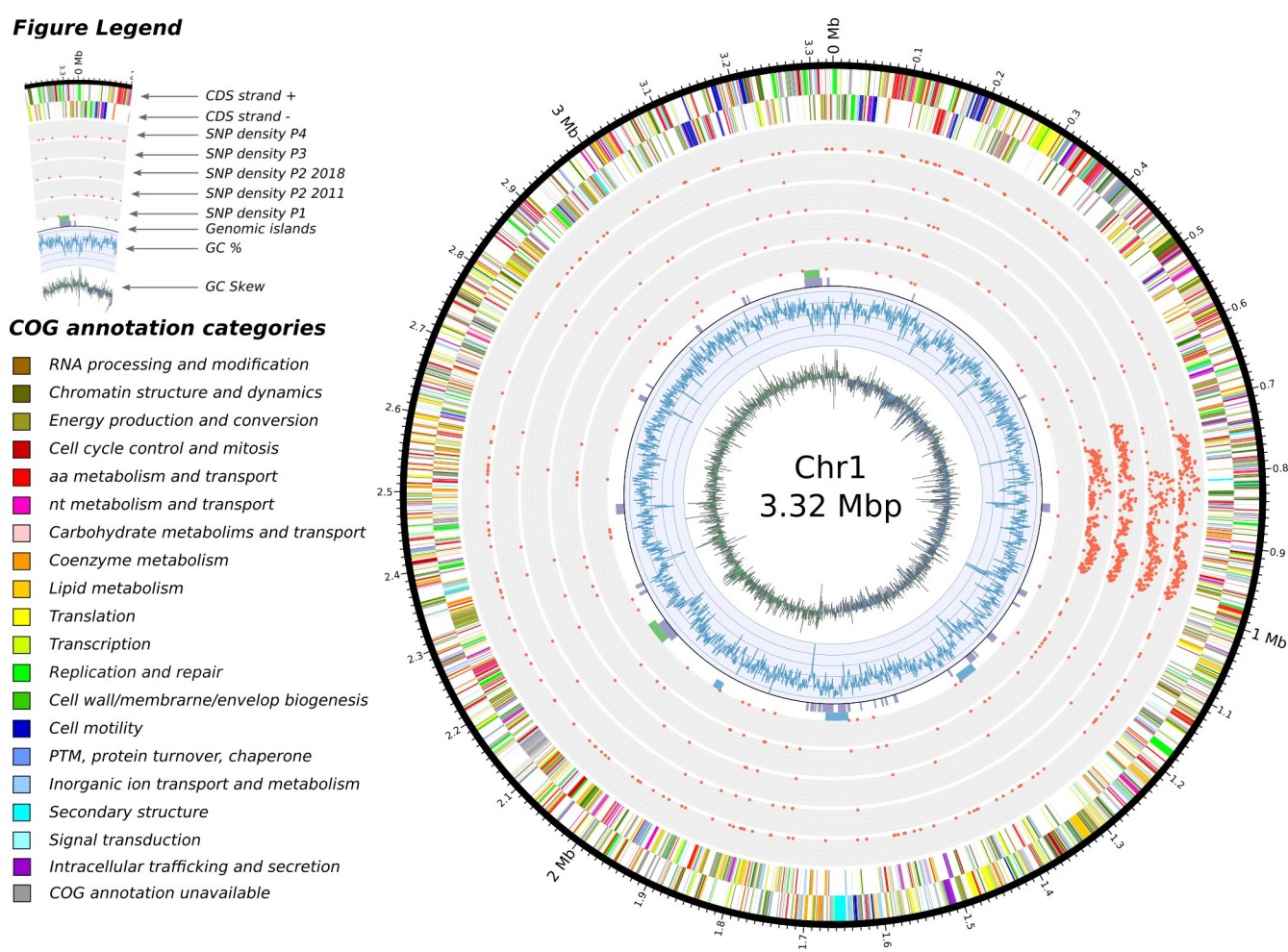

**Figure Legend**

- CDS strand +
- CDS strand -
- SNP density P4
- SNP density P3
- SNP density P2 2018
- SNP density P2 2011
- SNP density P1
- Genomic islands
- GC %
- GC Skew

**COG annotation categories**

- RNA processing and modification
- Chromatin structure and dynamics
- Energy production and conversion
- Cell cycle control and mitosis
- aa metabolism and transport
- nt metabolism and transport
- Carbohydrate metabolims and transport
- Coenzyme metabolism
- Lipid metabolism
- Translation
- Transcription
- Replication and repair
- Cell wall/membrarne/envelop biogenesis
- Cell motility
- PTM, protein turnover, chaperone
- Inorganic ion transport and metabolism
- Secondary structure
- Signal transduction
- Intracellular trafficking and secretion
- COG annotation unavailable

**Fig 5. Features of chromosome 1 and SNP densities comparisons.** In concentric circles, from the center and outward are depicted: 1) the GC skew, 2) the GC content, 3) the genomic islands detected by the software IslandViewer 4 (in purple), the prophages detected with Phaster (green) and the secondary metabolites identified by antiSMASH (in blue), 4) a total of five SNP densities, we separated the isolates from 2011 and the isolate from 2018 in patient 2 as we regard them as two separate acquisition events, 5) and 6) the annotated features found on the negative and positive strand respectively.

presence of many different *B. multivorans* STs in environmental reservoirs [22]. In contrast, between 2009 and 2019, ST-742 isolates were found in respiratory samples of four patients attending three different CF centers [26,28]. To investigate the epidemiology and evolution of these ST-742 isolates, we examined their genome sequences and their susceptibility to antimicrobial agents commonly used in CF therapy.

The high level of sequence identity, the collinearity of the replicons (S2 Fig), and the comparison with other publicly available *B. multivorans* genomes (Fig 4) demonstrated that all ST-742 isolates belonged to a single genomic lineage and that the shared ST was not simply a result of homoplasy [34,35]. This also corroborated an earlier study that showed that the ST predicted both phylogeny and gene content of *B. multivorans* isolates, supporting the use of MLST for epidemiological surveillance of Bcc bacteria [23].

The lack of evidence to support cross-infection [26] suggested that all patients acquired *B. multivorans* ST-742 from an environmental reservoir and that this strain is endemic in Belgium. A recent genomic study of *B. multivorans* reported that the same genomic lineages were isolated from CF and environmental samples and on different continents many years apart,

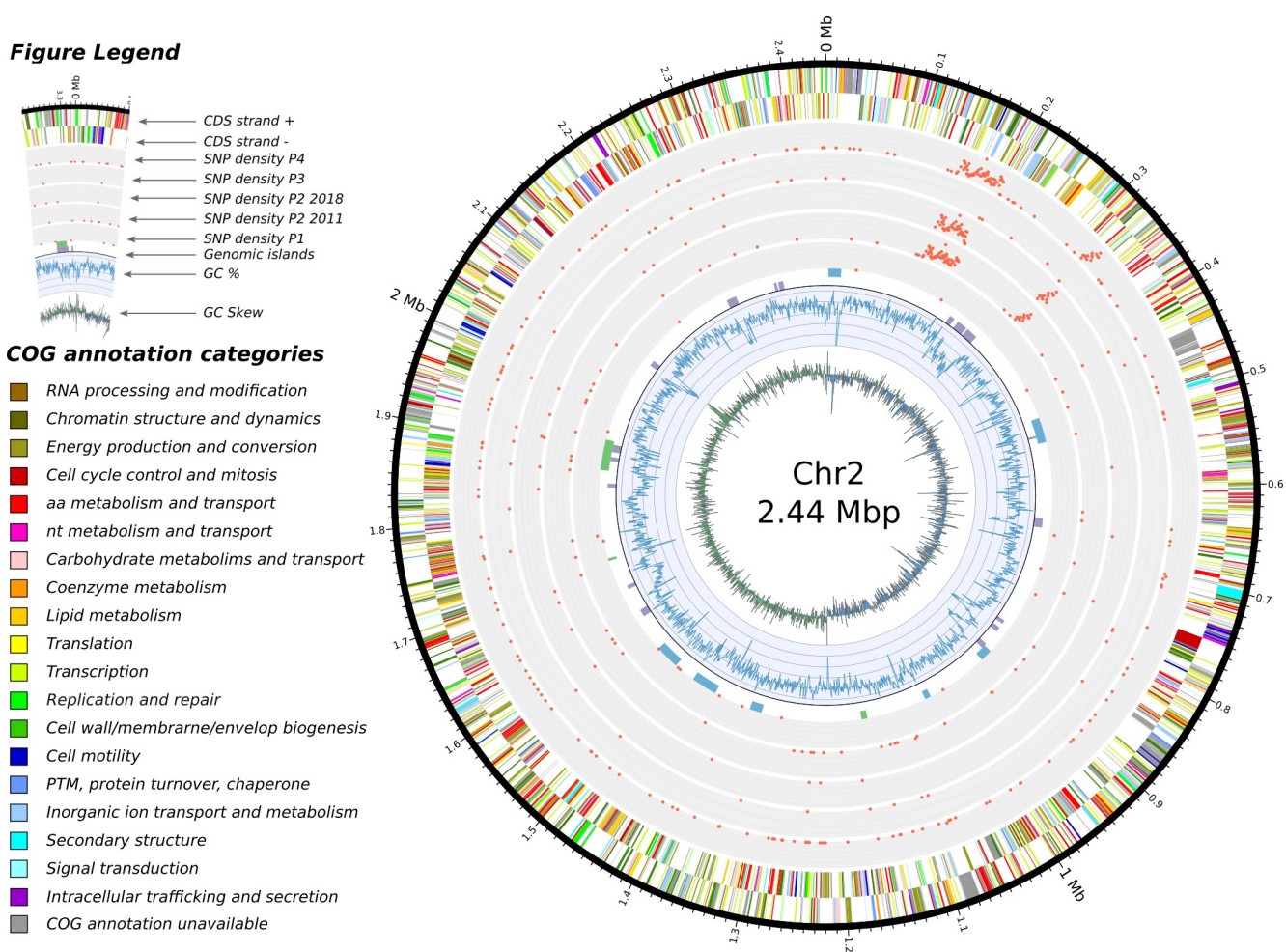

**Fig 6. Features of chromosome 2 and SNP densities comparisons.** In concentric circles, from the center and outward are depicted: 1) the GC skew, 2) the GC content, 3) the genomic islands detected by the software IslandViewer 4 (in purple), the prophages detected with Phaster (green) and the secondary metabolites identified by antiSMASH (in blue), 4) a total of five SNP densities, we separated the isolates from 2011 and the isolate from 2018 in patient 2 as we regard them as two separate acquisition events, 5) and 6) the annotated features found on the negative and positive strand respectively.

demonstrating the evolutionary persistence and ubiquity of these strains in different niches and on different continents [23]. Intriguingly, although several studies point towards the acquisition of *B. multivorans* from non-human sources such as the natural environment, its preferred natural habitat remains elusive [4,36].

Different elements suggest that this environmental reservoir holds a microdiverse *B. multivorans* ST-742 population [37] that is susceptible to several antimicrobial agents and from which different genetic variants colonized different patients. First, there is a considerable SNP diversity between genomes of isolates of different patients compared to the more limited diversity observed among the genomes of patient 1 isolates (Fig 4). Second, we observed different start positions of the high-density SNP regions in isolates from different patients (Figs 5, 6 and 7, Table 2). Third, we observed an apparent mobility of prophage elements which are differentially present in some isolates. Lastly, there was a shared plasmid present in the isolates of patient 2, 3 and 4 and a unique plasmid in the first isolate of patient 1 (Table A in S1 Tables). Indeed, this microdiverse *B. multivorans* ST-742 population appears to comprise different plasmids which can be dispensed once in a human host.

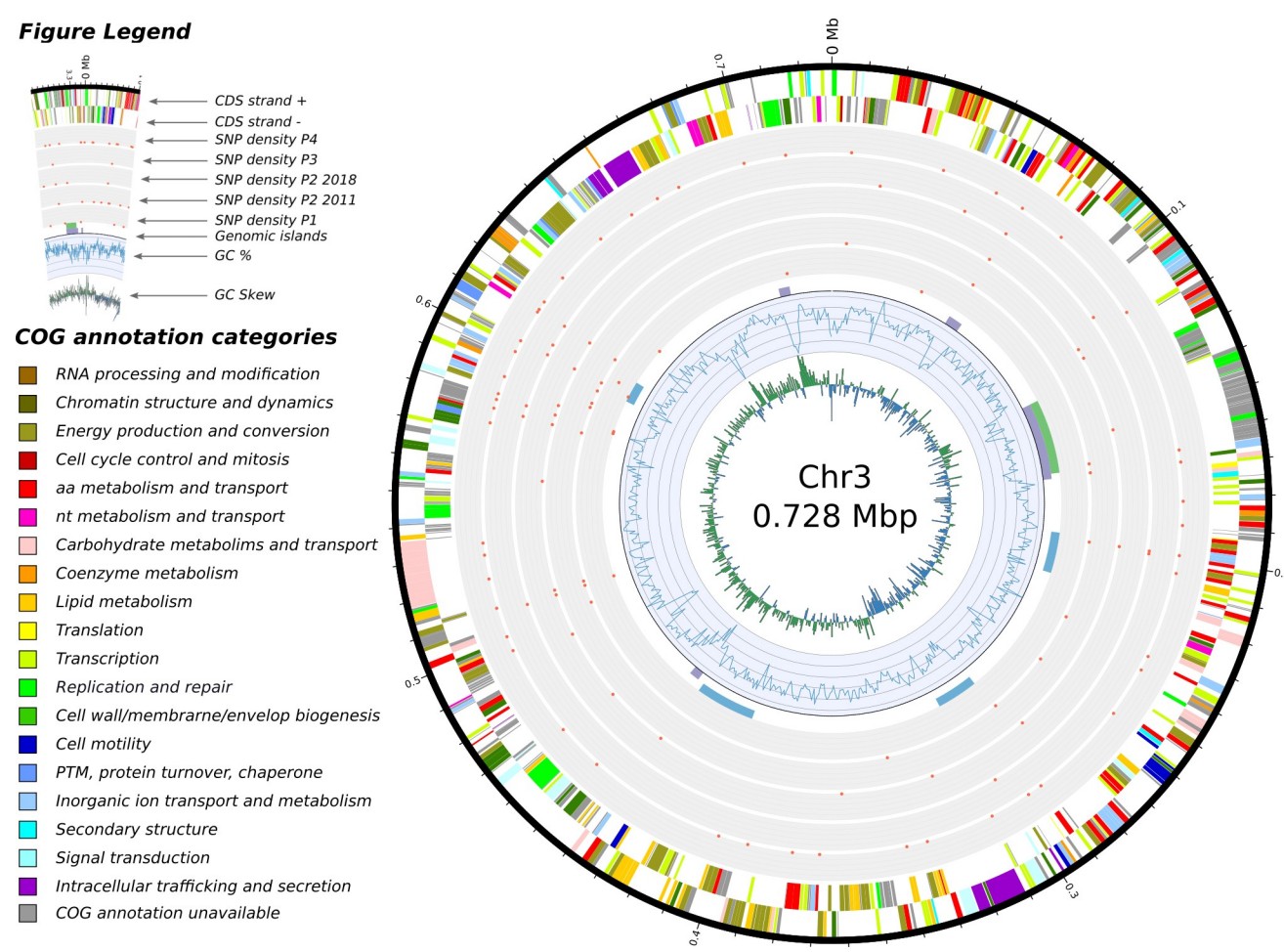

**Fig 7. Feature of chromosome 3 and SNP densities comparisons.** In concentric circles, from the center and outward are depicted: 1) the GC skew, 2) the GC content, 3) the genomic islands detected by the software IslandViewer 4 (in purple), the prophages detected with Phaster (green) and the secondary metabolites identified by antiSMASH (in blue), 4) a total of five SNP densities, we separated the isolates from 2011 and the isolate from 2018 in patient 2 as we regard them as two separate acquisition events, 5) and 6) the annotated features found on the negative and positive strand respectively.

De Boeck et al. [1] reported on the re-appearance of Bcc strains up to ten years after first colonization and hypothesized that reacquisition from the environment occurred. In patient 2, we hypothesize that isolate P2Bm2018 is derived from a re-infection from the same environmental reservoir and is not an immediate evolutionary descendant of the previously reported isolates of this patient (P2Bm2011a-b). This hypothesis is supported by the position of isolate P2Bm2018 in the core SNP tree (Fig 4), its different SNP profile (Figs 5, 6 and 7) with 1,990 unique SNPs when compared to P2Bm2011a-b, the presence of different regions with high SNP content in C1 and C2 (Table J in S1 Tables) and a differing pattern of prophage elements between early isolates (P2Bm2011a-b) and the late isolate (P2Bm2018). However, no plasmid was detected in P2Bm2018 and the antibiotic resistance profile of P2Bm2018 resembled that of P2Bm2011a (Fig 2), potentially indicating that the lineage of that isolate was already established in the lungs of the patient before 2018.

Comparative genomic analyses of Bcc isolates sampled from single patients in the course of chronic infections have shown that diversifying lineages can co-exist in the CF lungs for many years, with limited rates of evolution (as proxied by single nucleotide polymorphisms), i.e. around 2.1 to 2.4 SNPs/year [38–43]. This evolution within the lungs of CF patients can result

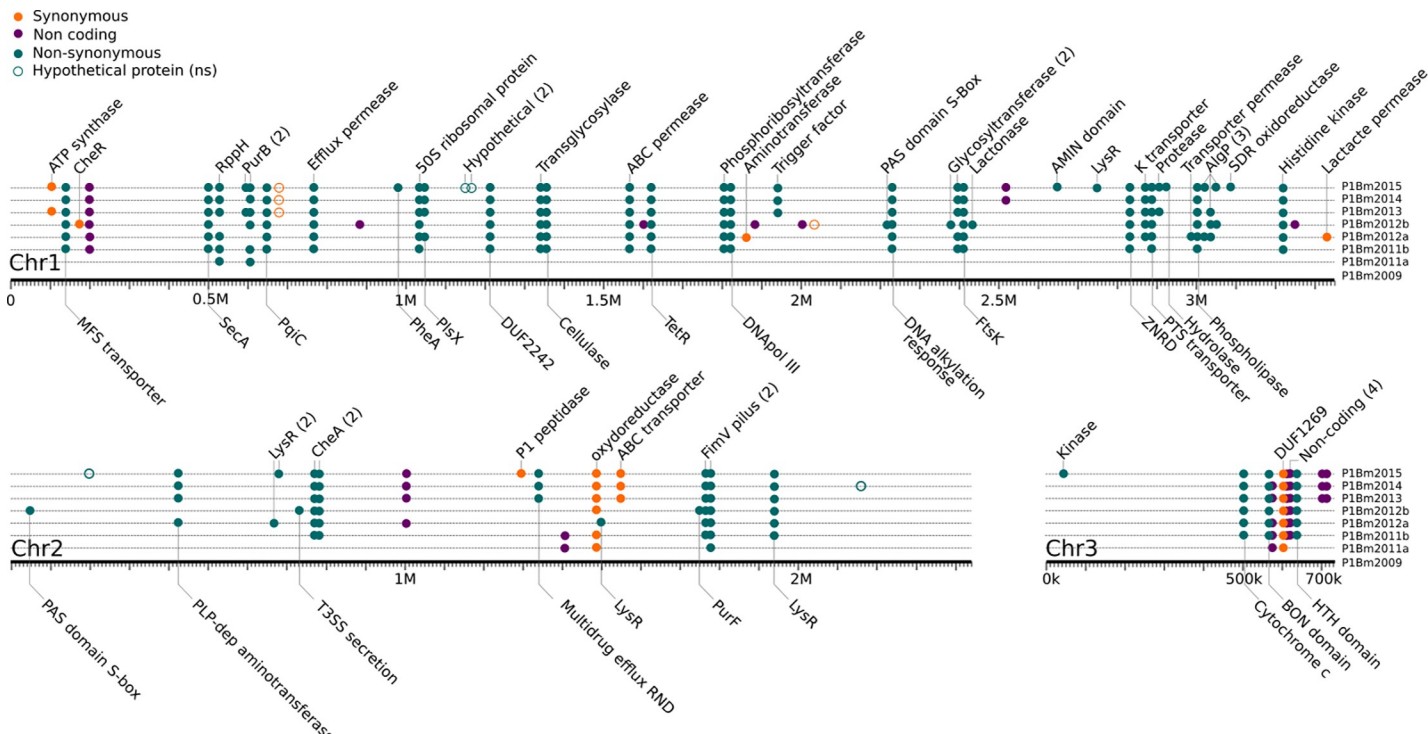

**Fig 8. Longitudinal evolution of isolates from patient 1.** Single nucleotide polymorphisms (SNPs) are shown for each of the three chromosomes of *B. multivorans*. When available, we additionally indicate the annotated function that is associated with the locus. The mutations that occur at multiple independent positions in the same locus, which are indicative of selective pressure, are marked with number between parentheses (Table 4).

in the loss of plasmids, increased antimicrobial resistance and a subset of genes that undergo positive selection [38,40,42–44].

In the present study, the presence of co-existing clades was supported in samples of patients 1 and 2, in which the same sample yielded isolates with distinct colony morphologies that showed distinct antibiotic resistance profiles (Fig 2, P2Bm2011a-b) and SNP profiles (Fig 8, P1Bm2011a-b and P1Bm2012a-b). Among the eight isolates of patient 1, no gradual increase in antimicrobial resistance was observed, as the isolates collected in 2011 were already fully resistant to virtually all compounds tested (Fig 2). We found an average mutation rate of 15 SNPs/year (S4 Fig) which was much higher than earlier reported values for *B. multivorans* (2.2–2.4 SNPs/year), *B. cenocepacia* (1.7–2.1 SNPs/year) or *B. dolosa* (2.1 SNPs/year) [39,40,42,43].

As expected from the evolution within the lungs of CF patients, we found a subset of genes with mutations at multiple independent positions (Table 3) and 30 loci that had a mutation in at least three patients (Table 4). As this latter set of genes underwent parallel adaptation in different patients, these genes may be involved in the adaptation of the ST-742 strain to the CF lung environment. Importantly, we sought to correlate the loci we discovered to those found in the *B. multivorans* literature. However, we found little commonalities despite almost all loci being present on the ST-742 strain (Table 3, Table L in S1 Tables). Indeed, only three mutations (i.e., ABC transporter permease, DNA-binding protein, and an acyltransferase) were found in common with the fixed mutations found in the study by Silva et al. (2016). Another three (hypothetical protein, LysR, and an amidophosphoribosyltransferase) mutations matched to the list of 61 pathoadapted proteins reported by Caballero et al. (2018), and a single

**Table 3. Loci with multiple independent mutations in patients 1 and 2.** Genes with multiple mutations in the corresponding CDSs of the longitudinal isolates from patient 1 (P1bm2009 –P1Bm2015) and patient 2 (P2Bm2011a, b), see Tables I and K in S1 Tables respectively. These adaptive mutations have not been reported in the *B. multivorans* literature (Table L in S1 Tables).

| Gene Product (locus) | Patient | Function | Previously identified |
|---|---|---|---|
| PurB (0592) | 1 | Adenylosuccinate lyase | no |
| Hypothetical (1166) | 1 | Fusaric acid resistance protein family | no |
| Glycosyltransferase (2375) | 1 | LPS modification | no |
| AlgP (2966) | 1 | Alginate production | no |
| LysR (3936) | 1 | Regulation of transcription | no |
| CheA (4025) | 1 | Chemotaxis | no |
| FimV (4911) | 1 | Type IV motility, pilus assembly | no |
| TetR (7680) | 2 | TetR family transcriptional regulator | no |
| Amidase (20235) | 2 | Amidase protein | no |

**Table 4. Loci undergoing parallel evolution in patients 1–4.** List of 30 locus with mutations (SNPs) in at least three patients (see also Table I in S1 Tables). We separated the isolates from 2011 and the isolate from 2018 in patient 2 as we regard them as two separate acquisition events.

| Mutated locus present? | | | | | Locus | Gene | Function |
|---|---|---|---|---|---|---|---|
| P1 | P2_2011 | P2_2018 | P3 | P4 | | | |
| ✓ | ✓ | | ✓ | ✓ | 0129 | | MFS transporter |
| ✓ | | ✓ | | ✓ | 1045 | plsX | phosphate acyltransferase PlsX |
| | ✓ | ✓ | ✓ | | 1134 | astD | succinylglutamate-semialdehyde dehydrogenase |
| | ✓ | ✓ | ✓ | | 1149 | | nodulation protein |
| ✓ | | ✓ | ✓ | ✓ | 1201 | | DUF2242 domain-containing protein |
| ✓ | | ✓ | ✓ | ✓ | 1600 | | TetR family transcriptional regulator |
| | ✓ | ✓ | ✓ | ✓ | 1764 | | protease modulator HflC |
| ✓ | | ✓ | ✓ | ✓ | 1788 | | DNA polymerase III subunit gamma/tau |
| ✓ | | ✓ | ✓ | ✓ | 2211 | | DNA alkylation response protein |
| | ✓ | ✓ | ✓ | ✓ | 2519 | | ATPase |
| | ✓ | ✓ | ✓ | | 2580 | | amino acid permease |
| ✓ | ✓ | | ✓ | ✓ | 2840 | | potassium transporter |
| ✓ | | ✓ | ✓ | ✓ | 2966 | | alginate biosynthesis protein AlgP |
| | ✓ | ✓ | ✓ | ✓ | 3147 | | hypothetical protein |
| | ✓ | ✓ | ✓ | ✓ | 3210 | rsmB | 16S rRNA (cytosine(967)-C(5))-methyltransferase |
| | ✓ | ✓ | ✓ | ✓ | 3483 | | MFS transporter |
| | ✓ | ✓ | ✓ | ✓ | 3702 | | GNAT family N-acetyltransferase |
| | ✓ | | ✓ | ✓ | 3720 | | hemagglutinin |
| | ✓ | | ✓ | ✓ | 4338 | | molybdopterin-dependent oxidoreductase |
| | ✓ | ✓ | ✓ | ✓ | 4442 | | CBS domain-containing protein |
| ✓ | | ✓ | ✓ | ✓ | 4911 | | pilus assembly protein FimV |
| | ✓ | ✓ | ✓ | ✓ | 5135 | | hypothetical protein |
| | ✓ | ✓ | ✓ | ✓ | 5336 | | sensor histidine kinase |
| | ✓ | ✓ | ✓ | ✓ | 5391 | | helix-turn-helix transcriptional regulator |
| | ✓ | ✓ | ✓ | ✓ | 5441 | | type II toxin-antitoxin system RelE/ParE |
| | ✓ | ✓ | ✓ | ✓ | 5799 | | ABC transporter substrate-binding protein |
| ✓ | ✓ | ✓ | ✓ | | 6056 | ctaD | cytochrome c oxidase subunit I |
| ✓ | | ✓ | ✓ | ✓ | 6124 | | BON domain-containing protein |
| ✓ | ✓ | ✓ | ✓ | ✓ | 6144 | | DUF1269 domain-containing protein |
| | ✓ | ✓ | ✓ | ✓ | 6156 | | LysR family transcriptional regulator |

ABC transporter in common with the recent study by Hassan et al. (2020). Only one gene (plsX, locus 1045) was previously reported by Silva et al. (2016) as part of adaptation within the lungs (Table 4, Table L in S1 Tables), while all other genes that underwent parallel evolution appear to be unique to this ST-742 lineage. This highlights the different strategies for adaptation that can be used by different strains and the crucial need for large-scale investigations similar to those conducted in *B. dolosa*, *B. cenocepacia* and *P. aeruginosa* [41,42,45]. This also corroborates the work of Hassan et al. (2020) who discussed that different strains appeared to have different routes of adaptation within the patient.

Using the short reads for SNP analysis, we observed that some regions of C1 and C2 had a remarkably high SNP density extending over 250 kb and 70 kb, respectively (Figs 5 and 6, Table J in S1 Tables). A previous study by Silva et al. (2016) hinted that potential hypermutator phenotypes existed in *B. multivorans* due to mutations in the DNA mismatch-repair mechanisms, specifically in MutL. However, we could not corroborate this hypothesis in our study. Although disruptive mutations are reported in the MutL gene of isolates from Patient 2 and 4 (SNP analysis against P1Bm2009, Table I in S1 Tables), the gene is found correctly annotated *de novo* in those isolates. Additionally, these regions were not identified as genomic islands, nor were they flanked by mobile elements. Functional analysis of their coding content did not reveal enrichments for specific COG categories when compared to the complete genome, which may indicate that the recombination of this region is tolerated by ST-742 (Table M in S1 Tables). The potential presence of these regions in other *B. multivorans* genomes (including both 97 public and 59 non-public sequences from the National Reference Centre database) was analyzed using BLASTn but no evidence was found that they originated from recombination between any of those strains. This suggested that they may represent events of recombination with donor strains for which no genome sequences are currently available. Additionally, as the start positions of the high-density SNP regions differ in isolates from different patients and share significant numbers of SNPs (Figs 5, 6 and 7 and S3, Table J in S1 Tables), these recombination events likely took place prior to infection, not during co-infection of the patients airways, and contribute to the microdiversity of the *B. multivorans* ST-742 population in its environmental reservoir [37,46].

Finally, from a technology perspective, we observed that while short read sequencing technologies such as Illumina were well suited to examine SNP diversity, long-read sequencing technologies readily allowed us to observe differences in plasmid and prophage content. In patient 2, we observed that prophages were mobile (Table 2), with an uncharacterized prophage element that was present on C1 of P2Bm2011b but absent from P2Bm2011a and P2Bm2018, and a second prophage, *Burkholderia* virus KS5 [47], that was present on C2 of P2Bm2011a and P2Bm2018, yet absent from P2Bm2011b. This could be relevant as previous studies in *P. aeruginosa* highlighted the importance of mobile prophage elements in driving within-host adaptation of the infecting strains [48]. Interestingly, we also noted two variants (deletions) in phosphoribosyltransferase and glycosyltransferase proteins that only appeared in the large structural variation analysis (Table 5) and corresponded to loci reported by Caballero et al. [38]. These findings illustrate the limitations of Illumina-based SNP analysis in providing a comprehensive list of genomic changes among isolates and emphasize the strength of combining sequencing technologies.

## Material and methods

### Ethics statement

The strains used in this study were collected by the participating clinical laboratories in the frame of routine diagnostic without additional testing and sent to the National Reference Center for *Burkholderia cepacia* complex where they were stored in the frame of activities mentioned

**Table 5. List of large structural variations in the isolates of Patient 1.** Using isolate P1Bm2009 as a reference, we mapped the long Nanopore reads of the subsequent isolates from Patient 1 to reveal large structural variations that escape SNP analysis. All structural variations are found in C1 and are fixed after their appearance in agreement with the SNP analysis (Fig 8). Isolates P1Bm2011a and P1Bm2012b, which originated from the same sputum sample as P1Bm2011b and P1Bm2012a, respectively, appear to come from separate lineages.

| Strains | Chr | Type | Function |
|---|---|---|---|
| P1Bm2013-2015 | 1 | Insertion | Intergenic, IS3 in the rRNA cluster (5S, 16S, 23S) |
| P1Bm2011b-2015 | 1 | Breakend | Glycosyltransferase |
| P1Bm2011b-2015 | 1 | Breakend | Intergenic, next to IS*256* |
| P1Bm2011a-2015 | 1 | Deletion | Phosphoribosyltransferase |
| P1Bm2012b | 1 | Deletion | Glycosyltransferase |
| P1Bm2015 | 1 | Deletion | SDR family NAD-p-dependent oxidoreductase |

in the Royal Decree 9 February 2011 setting the conditions for funding for reference centers in human microbiology and precised on the website of Sciensano: https://nrchm.wiv-isp.be/nl/oproep2019/legaal/default.aspx. In the frame of the specific terms of reference for the NRC for *Burkholderia cepacia* complex, a collection of representative strains was managed as outline in https://nrchm.wiv-isp.be/nl/oproep2019/lastenboek/Rapporten/STR06-Burkholderia%20cepacia_2019.pdf. Epidemiological data were collected retrospectively from patient charts and anonymously stored in a database. As no additional sampling or information was asked from patients, no formal approval from an ethical committee or informed consents are needed.

## Bacterial isolates and patient data

The thirteen clinical isolates presented in this study (Table 1) were isolated from sputum samples of four patients diagnosed with cystic fibrosis (Fig 1) and were collected at the Belgian National Reference Centre for *Burkholderia cepacia* complex (NRC Bcc). Isolates were identified as *Burkholderia multivorans* using MALDI-TOF mass spectrometry and *recA* gene sequence analysis [19,49]. Isolates were further molecularly characterized by multi-locus sequence typing (MLST) using the Bcc MLST scheme and the pubMLST database [19,21]. Isolates were grown aerobically in lysogeny broth (LB) medium at 37˚C and cultures were preserved in 25% glycerol at -80˚C.

For patient 1, *B. multivorans* was first detected in a sputum sample at age nine (P1Bm2009, January 2009). The initial treatment was intravenous (IV) ceftazidime, piperacillin-tazobactam and ciprofloxacin for 14 days, followed by inhaled temocillin and oral ciprofloxacin for 3 months. Cultures were negative until August 2009, date after which they remained positive until 2015, when the lung function declined drastically, and the patient received a double lung transplant. In 2011, two isolates from the same sputum sample were examined as they displayed a different colony morphology (P1Bm2011a and P1Bm2011b), and similarly in 2012 with isolates P1Bm2012a and P1Bm2012b (Fig 1).

For patient 2, *B. multivorans* was first detected in a sputum sample, at age five (December 2009, isolate not available for the NRC Bcc) and the infection is still ongoing at the time of the present study. Treatment has included IV administration of ceftazidime-amikacin, tobramycin, piperacillin, tazobactam, ciprofloxacin, and meropenem. Two isolates from the same sputum sample were sent to the NRC in 2012 as they displayed distinct colony morphologies (P2Bm2012a and P2Bm2012b), and an additional sample was received in 2018 (P2Bm2018) (Fig 1). Intermediate isolates (period 2012–2018) and later isolates (period 2019–2020) are not available for the NRC Bcc.

For patient 3, *B. multivorans* was first detected in a sputum sample at age nine (P3Bm2015, November 2015). The infection was successfully eradicated with an IV course of ceftazidime and tobramycin for 14 days, followed by 3 months of inhaled ceftazidime. No further samples were collected since.

For patient 4, *B. multivorans* was first detected in a sputum sample at age 25 (June 2017, not available for the NRC Bcc) and an isolate was sent to the NRC in December 2019 (P4bm2019) (Fig 1). The infection was treated for 14 days using intravenous meropenem, and no further therapy was administered due to other health related complications.

### Antibiotic sensitivity testing

We tested the *in vitro* susceptibility of all isolates against twenty different antibiotics (Table B in S1 Tables) as described previously [50]. The MIC values were determined by microdilutions in microtiter plates which were read on a Sensititre Vizion System device (ThermoScientific). To determine the *in vitro* susceptibility, EUCAST pharmacokinetic/pharmacodynamics (PK/PD) breakpoints were used, as no EUCAST species-specific breakpoints were available (tables available at https://www.eucast.org/clinical_breakpoints/ PK PD breakpoints).

### Genomic DNA isolation and hybrid sequencing

The isolates were inoculated in LB medium and grown overnight at 37°C to stationary phase ($OD_{600}$ over 1.0). The genomic DNA was extracted using the QIAGEN DNeasy ultraclean microbial kit. The quality and quantity of the extracted DNA was assessed using a Qubit 4.0 fluorometer, ThermoFisher Scientific Nanodrop spectrophotometer (OD280/260 and OD230/260), and agarose gel electrophoresis (1% w/v).

The genomic DNA was sequenced combining both short-read and long-read technologies. The first set of reads were obtained on an Illumina HiSeq4000 or NovaSeq6000 platform using a paired-end $2^*150$ bp approach at the Oxford Genomics Centre. The second set of reads were obtained on a MinION nanopore sequencer (Oxford Nanopore Technology) equipped with a flowcell of type R9.4.1 and a library prepared either with the 1D ligation approach or the Rapid library preparation kit (Table A in S1 Tables).

### Hybrid genome assembly

Quality of the Illumina reads was assessed using FastQC v0.11.9 [51] and Trimmomatic v0.38 [52] was used for adapter clipping, quality trimming (LEADING:3 TRAILING:3 SLIDING-WINDOW:4.15), and filtering on length (>50 bp). Quality of the Nanopore reads was assessed using Nanoplot v1.28.2 [53], while Porechop v0.2.3 [54] was used for barcode clipping and NanoFilt v2.6.0 [53] for filtering on quality (Q>8) and length (>500 bp).

The genomes of the isolates were assembled *de novo* using both the short-read SPAdes assembler v3.14.0 [55], and the hybrid assembler Unicycler v0.4.7 [56] with default options. The quality of the resulting assemblies was assessed using QUAST v5.0.2 [57] (S1 Fig) and the assembly graphs were visualized with Bandage v0.8.1 [58]. Pairwise average nucleotide identities between the ST-742 genomes were calculated using PYANI (v0.2.10) with the ANIb method.

### Functional annotation

All genomes were functionally annotated via the NCBI Prokaryotic Genome Annotation Pipeline [59]. The proteome was also annotated with COG categories and KEGG pathways using the eggNOG mapper [60]. This CDS-centric annotation was subsequently complemented by annotating genetic systems and genomic islands with 1) the prophage detection system PHASTER [61], 2) CRISPRCasFinder software [62], 3) genomic IslandViewer v4 [63], 4) ABRicate for the detection of acquired antimicrobial resistance genes [64], and 5) the biosynthetic gene cluster annotation tool ANTISmash v5 [65].

## Population structure and MLST

To examine the position of the clinical isolates within the general population structure of *B. multivorans*, all *B. multivorans* genomes were retrieved from the NCBI RefSeq database (release 98) as GenBank files (Table D in S1 Tables), together with *B. cenocepacia* J2315 [66] as an outgroup. Genomes with GC contents below 66% (one isolate), and more than 1,000 contigs (seven isolates) were excluded. Remaining GenBank files were converted into gff3 files using the perl script "bp_genbank2gff3" available in the BioPerl software library [67] and given as input to roary v3.1 3 [68] to delineate the pangenome and perform a core gene multiple sequence alignment. The core gene alignment was processed using FastTree v2.0 [69] to reconstruct the population structure. The phylogenetic tree was visualized and annotated using FigTree v1.4.4 [70]. MLST analysis on all downloaded genomes was performed using mlst (https://github.com/tseemann/mlst) and the Bcc pubMLST database [21].

## SNP analysis

SNP calling was done using snippy v.4.4.5 [71] with the quality-controlled Illumina reads of each isolate and the annotation of isolate P1Bm2009 or P2Bm2011a as reference. The SNP density was calculated by tabulating the number of unique SNPs per 1000 nucleotides windows along the genome (Figs 5, 6 and 7 and Table I in S1 Tables). The visualization of shared SNPs (S3 Fig) was created with the software UpsetR [72] and the Venn web application (http://bioinformatics.psb.ugent.be/webtools/Venn/). The core SNP alignment from snippy was used to construct a focused phylogenetic tree of our ST-742 isolates and the strain FDAARGOS_719 using FastTree v2.0 [69] and visualized using FigTree v1.4.4 [70]. Comparisons with previously published mutation data was conducted using BLASTp between the proteome of P1Bm2009 and the proteins reported in the studies listed in the Table L in S1 Tables.

## Large genomic structural variations

Overall sequence identity as well as collinearity of the genomes was examined by creating dot plot figures (S2 Fig) using Gepard v1.40 [73] with default options and word length adjusted to 100. The large structural variations were called using a combination of the long-read mapper ngmlr v0.2.7 [74] and the structural variation caller sniffles v1.0.11 [74] with the quality-controlled Nanopore reads and P1Bm2009 was as reference. The results were visually inspected using Ribbon v1.1 [75] and all the VCF files were consolidated into a spreadsheet, with the following filters applied: 1) variations related to the linearization of the replicons or smaller than 100 nt were removed, 2) imprecise variations were removed except when they matched a precise variations in another isolate, and 3) remaining variations in P1Bm2009 (three in total) were used to filter out false positives.

## Supporting information

**S1 Fig. Sequencing data, assembly metrics and assembly graphs.** The assembly graphs for the short-reads assemblies and hybrid assemblies are reported, enabling a quick visual inspection of the fragmentation of the assemblies as well as the presence of multiple replicons. (PNG)

**S2 Fig. Pairwise comparisons of the isolates chromosomes.** Chromosomes 1, 2, and 3 were compared in pairs using dot plots, highlighting the collinearity and overall sequence identity of the genomes across all isolates. Note that some chromosomes were not fully closed and subsequently rotated during the assembly resulting in visual artefacts including large inversions or re-arrangements. (PNG)

**S3 Fig. Shared and unique SNPs across the isolates from different patients.** This visualization supplements Figs 5, 6 and 7 by delineating the sharing of SNPs found in the different isolates. For each chromosome, we provide two visualizations of the data using UpSetR and a standard Venn diagram. We collapsed the 85 SNPs found in the isolates from P1Bm2009 to P1Bm2015 and kept separate the P2Bm2011a,b and P2Bm2018 isolates from patient 2, as they differed by 1,990 unique (not shared) SNPs.
(PNG)

**S4 Fig. Cumulative SNPs Patient 1.** The cumulative SNPs are plotted over the time period of infection in patient 1. A linear model was used to estimate the rate of appearance of SNPs (15 SNPs/year).
(TIF)

**S1 Tables. Table A: isolates description.** The thirteen ST-742 isolates are listed with a summary description of their sequencing and the related genomics data available the NCBI GenBank database. **Table B: antibiotic resistance profile.** The table contains the MIC values for all strains organized by type of antibiotics. The EUCAST or PK/PD interpretations are given by color (RED = resistant, ORANGE = intermediate, GREEN = sensitive). **Table C: location of genomic islands and genetic clusters.** Results from the three different software tools to predict the genomic locations of mobile elements and secondary metabolites clusters. **Table D: list of isolates from NCBI used for the population structure.** The genome sequences of the isolates were downloaded from the RefSeq database (release 98). **Table E: features of plasmid pP1Bm2009.** List of annotated features of plasmid 1 in patient 1 annotated using hmmer and eggNOG with a summary count of genes per COG categories. **Table F: features of plasmid pP2Bm2011a.** List of annotated features of plasmid 1 in patient 2 annotated using hmmer and eggNOG with a summary count of genes per COG categories and a comparison of COG categories between pP1Bm2009 and pP2Bm2011a. **Table G: master list of large structural variations.** List of all structural variations identified using the long reads (Nanopore data). **Table H: filtered list of large structural variations.** The filtering steps are described in the Materials and Methods section. **Table I: master list SNPs**. List of all SNPs identified when comparing the 13 isolates against P1Bm2009. **Table J: regions of high SNPs density**. Detailed start and stop positions of the regions displayed on the Figs 5, 6 and 7. **Table K: SNPs isolates patient 2**. List of SNPs identified when comparing the P2Bm2011b against P2Bm2011a. **Table L: List of loci under selection reported in previous studies.** This list contains the loci that were highlighted in previous longitudinal studies of *B. multivorans* infections of patients with cystic fibrosis. For each study, we mention the locus number (as reported), the NCBI protein accession number, whether or not the locus was present in the ST-742 strains, and whether there was a SNP found in the locus for the longitudinal isolates from patient 1. **Table M: COG annotation of the genes in high-density SNP regions.** Here we list the proteins found in the regions enriched in SNPs (Tables I and J in S1 Tables), functionally annotated with their COG categories. The graphic at the top compares the specific presence of the COG categories in those high-density SNP regions versus the complete genome.
(XLSX)

## Acknowledgments

We thank the referring laboratories of the following hospitals: Universitair Ziekenhuis Leuven (UZ Leuven; Leuven, Belgium), Clinique CHC MontLégia (Centre Hospitalier Chrétien, Liège, Belgium), and Universitair Ziekenhuis Brussel (UZ Brussel, Brussel, Belgium). We also thank

the Oxford Genomics Centre at the Welcome Centre for Human Genetics for the generation and initial processing of the Illumina sequencing data.

## Author Contributions

**Conceptualization:** Cédric Lood, Charlotte Peeters, Rob Lavigne, Peter Vandamme.

**Data curation:** Cédric Lood, Charlotte Peeters, Quentin Lamy-Besnier, Denis Piérard.

**Formal analysis:** Cédric Lood.

**Funding acquisition:** Vera van Noort, Rob Lavigne, Peter Vandamme.

**Investigation:** Cédric Lood, Charlotte Peeters, Quentin Lamy-Besnier, Jeroen Wagemans, Daniel De Vos, Marijke Proesmans, Jean-Paul Pirnay, Fedoua Echahidi, Denis Piérard, Matthieu Thimmesch, Anca Boeras, Katrien Lagrou, Evelien De Canck, Elke De Wachter, Rob Lavigne, Peter Vandamme.

**Methodology:** Cédric Lood, Charlotte Peeters, Fedoua Echahidi, Denis Piérard, Evelien De Canck.

**Project administration:** Cédric Lood, Charlotte Peeters.

**Resources:** Daniel De Vos, Marijke Proesmans, Jean-Paul Pirnay, Denis Piérard, Matthieu Thimmesch, Anca Boeras, Katrien Lagrou, Elke De Wachter, Rob Lavigne.

**Software:** Cédric Lood.

**Supervision:** Jeroen Wagemans, Daniel De Vos, Jean-Paul Pirnay, Denis Piérard, Vera van Noort, Rob Lavigne, Peter Vandamme.

**Validation:** Rob Lavigne, Peter Vandamme.

**Visualization:** Cédric Lood.

**Writing – original draft:** Cédric Lood, Charlotte Peeters.

**Writing – review & editing:** Cédric Lood, Charlotte Peeters, Quentin Lamy-Besnier, Jeroen Wagemans, Daniel De Vos, Marijke Proesmans, Jean-Paul Pirnay, Fedoua Echahidi, Denis Piérard, Matthieu Thimmesch, Anca Boeras, Katrien Lagrou, Evelien De Canck, Elke De Wachter, Vera van Noort, Rob Lavigne, Peter Vandamme.

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
