## [Decision Letter · Decision Letter 0]

11 Jan 2021

Dear Mr. Lood,

Thank you very much for submitting your manuscript "Genomics of an endemic cystic fibrosis Burkholderia multivorans strain reveals low within-patient evolution but high between-patient diversity" for consideration at PLOS Pathogens. As with all papers reviewed by the journal, your manuscript was reviewed by members of the editorial board and by several independent reviewers. The reviewers appreciated the attention to an important topic. Based on the reviews, we are likely to accept this manuscript for publication, providing that you modify the manuscript according to the review recommendations.

You will see that the three reviewers were generally positive. Two reviewers indicate they would like more in-depth and contextual interpretation of the data as presented. They have outlined their specific concerns/requests below. We agree with their assessment - and therefore we ask that you go through their comments and concerns as it will strengthen the impact of your work. However, given that most of the points are asking for more interpretation of your current analysis, as opposed to further experimentation or strong issues with the methods you employed, we believe you should be able to comply with these requests in a rather short amount of time.

Sincerely,

William Navarre

Associate Editor

PLOS Pathogens

Denise Monack

Section Editor

PLOS Pathogens

Kasturi Haldar

Editor-in-Chief

PLOS Pathogens

orcid.org/0000-0001-5065-158X

Michael Malim

Editor-in-Chief

PLOS Pathogens

orcid.org/0000-0002-7699-2064

You will see that the three reviewers were generally positive. Two reviewers indicate they would like more in-depth and contextual interpretation of the data as presented. They have outlined their specific concerns/requests below. We agree with their assessment and ask that you go through their comments and concerns as it will strengthen the impact of this work considerably. However, given that most of the points are asking for more interpretation of your current analysis, as opposed to further experimentation or strong issues with the methods you employed, we believe you should be able to comply with these requests in a rather short amount of time.

Reviewer Comments (if any, and for reference):

Reviewer's Responses to Questions

**Part I - Summary**

Reviewer #1: The manuscript by Lood et al describes a genomic analysis of Burkholderia multivorans isolates recovered from persons with cystic fibrosis (CF). The authors conclude that the findings demonstrate limited genomic variation among isolates serially recovered from the same person, but high diversity among isolates recovered from different persons.

In general, this is a well written, nicely presented manuscript that clearly describes comprehensive genomic analyses of an important opportunistic human pathogen. The work is of interest on a few levels: (i) the analysis of serial isolates from two individuals furthers our understanding of bacterial adaptation/diversification during chronic human infection; (ii) the study nicely illustrates the strength of integrated long- and sort-read WGS approaches to assess genomic differences between related bacteria; and (iii) the work provides convincing evidence of independent acquisition of the same strain of B multivorans by multiple persons with CF, which adds to our understanding of the epidemiology of Burkholderia and is an important consideration in infection control in CF. The analyses, including antibiotic susceptibility phenotype, phylogenetic context of this strain, and detailed SNP analyses support the authors conclusions.

Reviewer #2: Strengths

The manuscript "Genomics of an endemic cystic fibrosis Burkholderia multivorans strain reveals low within-patient evolution but high between-patient diversity" by Lood, Peeters, et al., is well-written, the data well-analyzed, and the conclusions provided are sound.

The combination of two different sequencing technologies is important, giving a broader picture of what is happening genetically, than either technique can produce alone.

It is of concern that the number of SNPs detected per year are significantly higher than that previously reported for Burkholderia cepacia complex strains. This suggests that even without being hypermutator strains, the strains analysed were capable of much higher rates of mutation than previously known.

The report highlights the importance of tracking the genetics of sequential patient isolates. Significantly, although the antibiotic resistances from strain to strain did not vary appreciably, that is not to say that there were not mutations occurring and being selected for. For each infecting strain, perhaps it is only a matter of time before the correct combination of random mutations coalesce to produce a better adapted variant, whether that be SNPs, plasmid loss, prophage acquisition, or some other genetic event.

Weaknesses

Although, as with any genomics study aimed at inferring past adaptations, this retrospective study cannot truly know that the mutations observed were "adaptive". Given that similar recent studies showed an accumulation of almost completely different SNPs, it suggests not only as the authors' state ... different strategies for adaptation to the human lung ..., but also possibly that the random mutations (or that all the mutations) are not significantly adaptive.

Reviewer #3: This is an exceptional comparative genomic analysis of a cluster of B. multivorans isolates belonging to a common lineage endemic to Belgium and capable of establishing chronic infections in persons with CF. The genomic methods are first-rate, attentive to structural variation and mobile elements, and show that this lineage is distinct from various others reported in this small field by undergoing patterns of adaptive evolution in vivo with limited overlap with prior studies. Another advantage is that four independent infections from this lineage are studied, enabling assessment of convergent evolution. Overall, this is a valuable contribution to the microbial genomic literature of Bcc, which remains relatively scarce.

Yet, the primary conclusion appears to be, as the abstract states, that these infections accumulate “peculiar patterns of genomic diversity between patients.” This leaves me with more questions than answers from this largely descriptive study, outlined below.

**Part II – Major Issues: Key Experiments Required for Acceptance**

Reviewer #1: I have no major concerns with any of the analyses and believe the conclusions are well-justified.

Reviewer #2: None.

Reviewer #3: 1. If they derived from a common, unknown environment, do the changes in vivo indicate shared adaptations that distinguish infected isolates from the natural source, in effect enabling “triangulation” of the source environment?

2. Is the natural environment a source for certain mobile elements that might be adaptive or maladaptive?

3. What are the phenotypic effects of any parallel genetic adaptations? How exactly (genetically or phenotypically) is this novel ST different from other infectious isolates in CF or environmental isolates from your prior work?

4. What can we learn about these dynamics about the epidemiology and evolution of opportunistic bacterial pathogens more generally? The manuscript offers little discussion outside our field of Bcc clinical microbiology, a missed opportunity.

5. Given that a major discovery is of clusters of mutations in certain regions, which as the authors acknowledge suggest recombination events, what is the novelty? That recombination may happen perhaps due to coinfection in the airway or possibly in the reservoir prior to reinfection?

**Part III – Minor Issues: Editorial and Data Presentation Modifications**

Reviewer #1: Lines 39 and 291: SNP is typically used to describe single nucleotide polymorphisms (not small nucleotide…)

Line 53: necrosis, not necrotisis

Line 70: “…higher RATES OF morbidity and mortality…”

Line 105: omit “in”

Line 168: define ‘indels’ with first use

Line 338: elements, not element

Line 348: omit “different”

Methods: check spelling of antibiotics throughout: be consistent, use English spellings

Line 385: can a reference be provided for EUCAST PK/PD breakpoints?

Line 697: each originated from, instead of “were originating from”

Figures 3, 5-7: can these be uploaded in higher quality; very difficult to read even with magnification

Reviewer #2: Minor issues

Line 50: In the author summary, the authors (surprisingly) indicate that " multivorans is the most prevalent species within the Burkholderia cepacia complex (Bcc) found infecting the lungs of patients with cystic fibrosis (CF)." Later, in lines 96 and 244, they go on to clarify that this fact is true for Belgium. In fact, although changes have occurred in some clinics and some countries recently, there are clinics and countries where B. cenocepacia is still the most prevalent Bcc species. Therefore, the authors should change the wording on line 50 to say "In some countires, B. multivorans is the most prevalent species ....".

Line 53: the word "necrotisis is rarely used. The more highly preferred English word is "necrosis".

Line 151: The authors refer to the "typical" conserved genome structure of Bm being three large replicons. If this structure is typical, then they are referring to previously determined data, and therefore should supply references for this work.

Line 316: Or perhaps elsewhere. The authors indicate the locations of high SNP density in some of the replicons in some of the strains. The authors should comment on these stretches further. Why is this happening, What genes in common are being affected if any, and Why here and not everywhere. (Do these modified regions suggest improvements in adaptation, or merely tolerable unimportant mutations?)

Line 416: It is apparent that PHAST can identify additional prophage element components not detected by PHASTER. Did the authors attempt to use this, or any of the other recent prophage software packages (e.g. ProphET, Prophage Hunter, PhageWeb, Phage_Finder, PhiSpy, LysoPhD, ) to analyze their data?

Reviewer #3: (No Response)

PLOS authors have the option to publish the peer review history of their article (what does this mean?). If published, this will include your full peer review and any attached files.

Reviewer #1: **Yes: **John J. LiPuma

Reviewer #2: No

Reviewer #3: **Yes: **Vaughn Cooper
---

## [Editor Report · Decision Letter 1]

22 Feb 2021

Dear Mr. Lood,

We are pleased to inform you that your manuscript 'Genomics of an endemic cystic fibrosis Burkholderia multivorans strain reveals low within-patient evolution but high between-patient diversity' has been provisionally accepted for publication in PLOS Pathogens.

Best regards,

William Navarre

Associate Editor

PLOS Pathogens

Denise Monack

Section Editor

PLOS Pathogens

Kasturi Haldar

Editor-in-Chief

PLOS Pathogens

orcid.org/0000-0001-5065-158X

Michael Malim

Editor-in-Chief

PLOS Pathogens

orcid.org/0000-0002-7699-2064

We note that all three reviewers were generally supportive of publication during the last round of review.  The associate editor has reviewed the recent changes to the manuscript and believes the concerns by the reviewers have been adequately addressed.

---

## [Editor Report · Acceptance letter]

9 Mar 2021

Dear Dr. Lavigne,

We are delighted to inform you that your manuscript, "Genomics of an endemic cystic fibrosis *Burkholderia multivorans* strain reveals low within-patient evolution but high between-patient diversity," has been formally accepted for publication in PLOS Pathogens.

Best regards,

Kasturi Haldar

Editor-in-Chief

PLOS Pathogens

orcid.org/0000-0001-5065-158X

Michael Malim

Editor-in-Chief

PLOS Pathogens

orcid.org/0000-0002-7699-2064